# Host Factors in Dysregulation of the Gut Barrier Function during Alcohol-Associated Liver Disease

**DOI:** 10.3390/ijms222312687

**Published:** 2021-11-24

**Authors:** Luca Maccioni, Isabelle A. Leclercq, Bernd Schnabl, Peter Stärkel

**Affiliations:** 1Laboratory of Hepato-Gastroenterology, Institute of Experimental and Clinical Research, UCLouvain, Unversité Catholique de Louvain, 1200 Brussels, Belgium; luca.maccioni@uclouvain.be (L.M.); isabelle.leclercq@uclouvain.be (I.A.L.); 2Department of Medicine, University of California San Diego, La Jolla, CA 92093, USA; beschnabl@health.ucsd.edu; 3Department of Medicine, VA San Diego Healthcare System, San Diego, CA 92161, USA; 4Department of Hepato-Gastroenterology, Cliniques Universitaires Saint-Luc, 1200 Brussels, Belgium

**Keywords:** intestinal barrier, chronic alcohol consumption, gut-liver axis, intestinal immunity, inflammation

## Abstract

Chronic alcohol consumption and alcohol-associated liver disease (ALD) represent a major public health problem worldwide. Only a minority of patients with an alcohol-use disorder (AUD) develop severe forms of liver disease (e.g., steatohepatitis and fibrosis) and finally progress to the more advanced stages of ALD, such as severe alcohol-associated hepatitis and decompensated cirrhosis. Emerging evidence suggests that gut barrier dysfunction is multifactorial, implicating microbiota changes, alterations in the intestinal epithelium, and immune dysfunction. This failing gut barrier ultimately allows microbial antigens, microbes, and metabolites to translocate to the liver and into systemic circulation. Subsequent activation of immune and inflammatory responses contributes to liver disease progression. Here we review the literature about the disturbance of the different host defense mechanisms linked to gut barrier dysfunction, increased microbial translocation, and impairment of liver and systemic inflammatory responses in the different stages of ALD.

## 1. Introduction

Chronic alcohol consumption is one of the leading causes of chronic liver disease and liver-related deaths worldwide. Although more than 90–95% of people with excessive alcohol drinking develop fatty liver, only a minority (10–20%) of subjects with an alcohol-use disorder (AUD) ultimately progress to more advanced stages of alcohol-associated liver disease (ALD) (e.g., severe alcohol-associated hepatitis and decompensated cirrhosis) and its related complications [1]. Some factors that are correlated with disease progression have been identified, including sex, drinking patterns, genetics, obesity, and dietary factors. Those factors either increase susceptibility to liver damage or aggravate disease by acting synergistically with alcohol [2,3]. The interaction between obesity and alcohol, in particular for mild or moderate consumption, is controversial in humans [4]. The incidence of obesity and the associated metabolic syndrome is increasing worldwide. A recent cohort study showed that patients with excessive alcohol consumption have increased prevalence of obesity and metabolic syndrome [5], which might increase the risk for liver disease progression. It has been reported that moderate alcohol consumption increases the risk of development of hepatic steatosis in association with obesity. In contrast, moderate alcohol consumption reduces mortality in normal body mass index (BMI) and overweight individuals. This beneficial effect was, however, lost in obese subjects [6]. In rodents, the synergism between alcohol and obesity might be linked to alterations in immune responses, metabolism, and gut microbiota composition [3]. Recent animal data where alcohol binges have been added on top of a non-alcoholic steatohepatitis model support a synergistic effect for liver damage (for a detailed review see [7]). Even if some advances have been made in our understanding of the factors involved in the development of ALD, there are still no FDA-approved therapies for treating patients with ALD and abstinence is considered the most effective treatment option [2]. Drugs aiming at reducing the immune responses have been studied for patients with severe forms of liver disease, especially in severe alcohol-associated hepatitis [2]. However, none of these approaches have proven to be safe and efficient [2].

The pathophysiology of gut barrier dysfunction associated with alcohol abuse is likely multifactorial. Changes in the intestinal microbiome, including bacteria, fungi, and viruses, have been associated with ALD [8,9,10,11]. This so-called intestinal dysbiosis together with increased intestinal permeability is thought to lead to microbial translocation. Microbes and/or pathogen-associated molecular patterns (PAMPs) reach the liver and serve as a trigger for the initiation of an inflammatory response. Consequently, studies targeted the gut microbes as part of the intestinal barrier for the potential treatment of ALD with some success (for a more detailed review see [12]). Other reports have described how targeting intestinal permeability might prove beneficial also as a potential treatment for liver diseases [13].

This general concept primarily derived from animal studies is, however, challenged by observations made in humans indicating that only approximately 40% of subjects with alcohol abuse have measurable intestinal dysbiosis [14]. Moreover, increased intestinal permeability in humans does not seem to be a prerequisite for microbial translocation to occur [15]. These observations imply that additional factors beyond microbiota changes and intestinal permeability are likely implicated in the pathophysiology of ALD. They also highlight the difficulties with extrapolating animal data to human pathology for several reasons: (1) animals have a natural aversion to alcohol, requiring ways of administration that do not mimic human alcohol-seeking behavior [16,17]; (2) rodents have a 5-times faster ethanol metabolism, making it difficult to reach stable high alcohol levels [18]; (3) compared to humans, rodents are characterized by profound differences in their immune system [19] and their microbiota [20]; and (4) animals only develop mild forms of ALD upon chronic alcohol feeding and even the most “sophisticated models” do not resume the liver-damage pattern observed in humans [21]. 

Currently, detailed mechanisms of gut barrier dysfunction, especially in humans, are not available, and we generally lack a cause–effect relationship between changes in the intestine and ALD progression. Most of the available data is based on a series that have included only small numbers of patients which do not allow to consider heterogeneity and potential high variability in humans. The functions of the gut barrier are also ensured by factors that are specific to the host, such as the mucus, antimicrobial molecules, and immune cells. Emerging evidence suggests that dynamic modifications in the intestinal epithelium and immune responses as well as reduced immunosurveillance of microbes occur in parallel with microbial translocation and are associated with liver disease severity [22]. Because of the interaction between the microbiota and the host it is not entirely possible to dichotomize ALD pathophysiology into microbial changes and alterations in the host. However, a comprehensive overview of the host-related intestinal alterations linked to the different stages of ALD is lacking. Therefore, this review aims to highlight the advances made in the comprehension of the host factors involved in the maintenance of the intestinal barrier in healthy conditions and the changes linked to alcohol abuse and ALD progression, with specific attention paid to the data described from humans.

## 2. The Gut Barrier

### 2.1. Gut Microbiome

The gut is the natural habitat for 100 trillion microorganisms (bacteria, archaea, fungi, and viruses) [23]. This community of microbe’s genome—the microbiome—encodes 100 times more genes than the human genome [24]. These microbes exert a remarkable influence on the host during homeostasis and disease. Indeed, the gut microbiota can modulate multiple physiological functions of the host, such as strengthening gut integrity and shaping the intestinal epithelium, harvesting energy [25], protecting against pathogens [26], and regulating immune response [27]. 

The gut microbiome represents our first line of defense against potential pathogens [28]. Intestinal microbes compete with pathogens for nutrients and produce antimicrobial molecules called bacteriocins [29], which shape the composition of the gut microbiota. Commensal microbes further contribute to the development of the mucosal immune system via direct interactions with host epithelial and immune cells [30]. In addition, the gut microbiome is responsible for a number of vital metabolic and signaling functions for the host, such as biotransformation of primary bile acids [31], production of short-chain fatty acids [25], synthesis of all B vitamins and vitamin K [32], as well as synthesis of essential amino acids [33].

### 2.2. Intestinal Epithelium

The principal functions of the intestinal barrier are nutrient absorption and defense against invading pathogens and potentially harmful substances. Mucosal cell types dynamically contribute to a balance between protective immunity and prevention of excessive immune responses against nutrients and commensal microbes. A first single-layer of cells, called the intestinal epithelium, facilitate absorption of nutrients while preventing microbial attachment and their translocation into the blood circulation [34]. 

The small intestinal epithelium is organized in villi and crypts of Lieberkühn. Adult stem cells reside at the base of the crypts and give rise to all absorptive and secretory cells making up the epithelial layer [35]. Gut barrier functions are insured by different epithelial cell types. Secretory goblet cells produce mucus—a viscous fluid enriched in mucin glycoproteins that form a first physical barrier against luminal microbes. Commensal bacteria also degrade glycans to extract the energy content, which then they share with the host in a mutualistic relationship [36]. Moreover, goblet cells produce antimicrobial peptides, such as trefoil factor [37] and resistin-like molecule beta [38], to dampen the mucosal attachment of microbes. In addition, they are also able to present bacterial antigens to dendritic cells via the so-called “goblet cells antigen-associated passages” (GAPs), thus educating the intestinal immune system [39]. Paneth cells, which reside at the base of the crypts, contribute to gut barrier function by producing and releasing into the crypt lumen antimicrobial molecules such as antimicrobial peptides, the regenerating islet-derived 3 (Reg3) family of proteins, lysozyme, and secretory phospholipase A2 [40]. Microfold cells, also known as M cells, are located in the follicle-associated epithelium. They uptake antigens and microorganisms from the lumen and deliver to organized lymphoid tissues within the mucosa [41]. Absorptive enterocytes also contribute to defense mechanisms by regulating paracellular permeability with tight junction proteins and directly recognizing a variety of PAMPs with different pattern recognition receptors, including Toll-like receptors (TLRs) and nucleotide-binding oligomerization domain-containing proteins (NODs) [42]. Furthermore, a dense glycocalyx consisting of extended and heavily glycosylated membrane mucins covers the surface of enterocytes, reinforcing the protective properties of the intestinal mucus [43].

### 2.3. Intestinal Immune System

The mucosal immune system plays an important role in orchestrating intestinal defense mechanisms against potential pathogenic microorganisms by coordinating innate and adaptive responses. Immune cells strategically localize in different compartments of the gut mucosa according to the expression of different receptors and the capacity to sense directly or indirectly microbes or their products. 

Macrophages are in part found under the epithelium in the lamina propria where they engulf and kill invading microorganisms and subsequently expose their antigens to adaptive immune cells. Unlike phagocytes in other parts of the body, gut macrophages produce large amounts of regulatory cytokines, such as IL-10, and contribute to maintain a tolerant status in the intestine while avoiding excessive inflammatory reactions [44]. The intestinal macrophages are continuously replaced by circulating monocytes. However, a recent report sheds light on the existence of long-lived tissue-resident gut macrophages that are not replaced by infiltrating monocytes [45]. 

Dendritic cells (DCs) orchestrate mucosal immunity by acquiring antigens in the epithelium through goblet or microfold epithelial cells or by directly capturing and sampling luminal antigens. They then processed antigens to T cells and B cells in the mucosa or in organized secondary lymphoid structures known as Peyer’s Patches, as well as in mesenteric lymph nodes [46]. Tolerogenic dendritic cells promote differentiation of regulatory T cells (Treg) [47], which in turn are able to induce tolerogenic DCs via IL-10, TGF-β, and other surface molecules [48]. DCs in gut-associated lymphoid tissue (GALT) and mesenteric lymph nodes further promote differentiation of B cells into immunoglobulin A (IgA)-producing plasma cells and memory B cells. Dimers of immunoglobulin A are transported through intestinal epithelial cells and secreted into the lumen where it binds commensal and pathogenic microbes, without initiating a pro-inflammatory response [49], and therefore contributes to mucosal defense. 

Recent investigations have revealed the existence of CD8+ and CD4+ T resident memory cells (TRM) and their importance in the maintenance of gut microbial immune surveillance. These lymphocytes are localized in the intestinal epithelium and lamina propria and allow the fast initiation of targeted immune responses against potential invaders [50] (Figure 1A).

## 3. Gut Barrier Dysfunction in the Context of Chronic Alcohol Consumption

Alcohol abuse is associated with small intestinal bacterial overgrowth [51], alterations in the composition of the intestinal micro- and mycobiota, also called as gut dysbiosis [8,52], and increased microbial translocation [53]. In addition, both experimental and clinical evidence show modifications at multiple levels of gut barrier function, including the epithelium and immune system, which might shape the microbial changes, allow microbes to translocate to the portal circulation, and eventually participate in alcohol-associated liver disease progression. We here focus on the dysfunctional defense mechanisms in the different compartments and cell types that finally contribute to the overall failure of the gut barrier.

### 3.1. Alcohol-Associated Changes in the Gut Microbiome

Most of the studies have investigated the changes in the microbiota composition looking specifically at bacteria. Chronic alcohol consumption can lead to intestinal bacterial overgrowth [51], compositional changes in the bacterial microbiota [54], and elevated systemic levels of bacterial-derived products [53]. Alterations in the bacterial microbiota and their associations with chronic alcohol consumption in humans have been reviewed in more detail in [55].

Recent reports indicate alcohol-associated overgrowth of gut fungal populations [56], with reduced fungal diversity and a proportional increase in *Candida albicans* (*C. albicans*) and *Malassezia restricta* (*M. restricta*) in patients with alcohol-use disorder [57]. Regarding the enteric viruses, only one study by Jiang et al. has evaluated the fecal virome of patients with ALD at different disease stage (e.g., severe alcohol-associated hepatitis, alcohol-use disorder) in comparison to healthy controls [11]. Patients with severe alcohol-associated hepatitis were characterized by a marked increase in mammalian viruses, such as Parvoviridae and Herpesviridae [11]. In addition, many bacteriophages were more abundant in patients with severe alcohol-associated hepatitis, including *Staphylococcus*, *Escherichia*, Enterobacteria, and *Enterococcus* phages [11].

Gut dysbiosis is likely one factor that contributes to increased microbial translocation, either directly or in an indirect manner. Overrepresented pathogenic microbes can attach and invade the host by yet not completely elucidated mechanisms. Moreover, loss of commensal microbes involved in different metabolic functions (e.g., transformation of bile acids, synthesis of vitamins, short-chain fatty acids and amino acids) can negatively affect the intestinal epithelium and immune system [30].

### 3.2. Alcohol-Associated Changes in the Intestinal Epithelium

Several independent studies have shown alcohol-induced disruption of tight junction proteins (e.g., zonula occludens 1 (ZO-1), occludin, and claudins) and subsequent increased paracellular permeability, also called leaky gut, primarily confined to the proximal small intestine (for a more detailed review, see [58]). However, a leaky gut was observed in only a fraction of AUD patients at early ALD stages (steatohepatitis, steato-fibrosis) [14,15]. In contrast to animal models and in presence of more advanced stages of liver disease, such as decompensated cirrhosis, where additional factors (e.g., Portal hypertension) lead to increased paracellular permeability [55], we found that leaky gut alone could not explain increased systemic microbial translocation at these early disease stages in AUD patients [15]. These observations indicate that other defense mechanisms need to fail in order to allow microbes and/or their products to cross the intestinal barrier. 

Both animal models and subjects with an alcohol-use disorder are characterized by increased production of mucus in the proximal small intestine [59]. Mucin-2 deficiency in mice leads to amelioration of alcohol-induced liver disease. The protective effect was demonstrated to be dependent on changes in the composition of the microbiota and increased expression of Reg3 antimicrobial molecules, resulting in reduced bacterial overload and translocation in mice [59]. Moreover, glycosylation of the mucins in the intestine is impaired since Fut2, an enzyme involved in α1–2-fucosylation at the cell surface of enterocytes and mucus [60], is downregulated in AUD patients [61]. Membrane and secreted α1–2-linked fucose can be cleaved by bacterial fucosidase, and the liberated L-fucose is utilized by certain bacteria [62]. Fut2 deficiency could therefore induce overgrowth of potential pathogens in the mucosa. Deficiency of Fut2 exacerbates chronic ethanol-induced liver injury, steatosis, and inflammation in mice [61], supporting its protective role against alcohol-induced liver disease.

Chronic alcohol consumption reduces gene and protein expression of Reg3γ and Reg3β in murine small intestine and duodenal Reg3γ in AUD patients, indicating impaired function of specialized Paneth cells [54]. Deficiency in Reg3 lectins in mice enhances bacterial translocation into the liver with subsequent activation of the immune responses [54]. Furthermore, Reg3 knockout mice chronically exposed to ethanol had worsened liver injury linked to increased bacterial translocation [51]. Interestingly, diminution of the antimicrobial molecules Reg3 and Human defensin 5 (own unpublished data) are associated with increased number of mucosa-associated bacteria [51] and increased systemic microbial translocation in actively drinking subjects compared to matched healthy subjects [15]. 

The maintenance of a correct balance between proliferation and differentiation programs is crucial for functions of the gut barrier, such as correct absorption of nutrients and coordination of defense mechanisms against invading microbes. Mice chronically fed ethanol are characterized by damage of adult stem cells via dysregulation of Wnt/β–Catenin signaling and reduced expression of stem cell markers such as Lgr5 and Bmi1 [63]. In contrast, animal models of chronic and binge alcohol consumption show crypt hyperplasia (increased crypt length) with enhanced proliferation through the Wnt/β–Catenin and Erk1/2 signaling pathways [64]. Duodenal biopsies in actively drinking subjects also revealed reduced villi length [15] and increased crypt hyperplasia (own unpublished data). The causes of the alterations in the small intestinal epithelium in humans are still incompletely understood. They might be related to modifications in the proliferation/differentiation program, which remain to be elucidated. Epithelial alterations could influence both defective defense mechanisms as well as malabsorption of nutrients, a well-known characteristic of patients with an alcohol-use disorder [65].

### 3.3. Alcohol-Induced Alterations in the Intestinal Immune System

Chronic alcohol consumption is associated with impairment of both innate and adaptive immune responses. How these alterations affect attachment and invasion of microbes in the gut mucosa in the different stages of ALD is still incompletely understood.

#### 3.3.1. Macrophages

Recent finding from our laboratory showed a reduced number of macrophages in the duodenum of AUD patients associated with elevated microbial translocation [15]. Whether this reduced number of intestinal macrophages is due to a decrease in resident macrophages or of infiltrating monocytes is still not known. Macrophages in the lamina propria that produce TNF-α are increased in the proximal small intestine of mice chronically fed ethanol as well as in AUD patients [66]. The pro- (IL1β) and anti-inflammatory cytokines related to macrophages (IL-10 and TGFβ) are upregulated in distal duodenal biopsies from actively drinking subjects compared to the controls [15]. Which subsets of macrophages produce these inflammatory mediators with opposite roles (inflammatory vs. tolerogenic) remain unknown as well as its significance for defense mechanisms. Whether and how these changes relate to ALD also requires further investigation.

#### 3.3.2. Dendritic Cells

Dendritic cells, together with macrophages, orchestrate immune responses in the gut barrier through presentation of the processed antigens to adaptive immune cells. Few studies have investigated the impact of alcohol abuse on dendritic cells. Acute alcohol intake alters the function and cytokine production of human monocyte-derived DCs in the blood circulation [67]. Moreover, chronic alcohol consumption is associated with changes in the distribution, immunophenotype, and secretion of inflammatory mediators [68]. Rhesus macaques chronically exposed to ethanol show a reduced number of bone marrow and circulating pools of myeloid dendritic cells, with suppression of costimulatory molecule such as CD83, which may attenuate the ability of DCs to promote T cell expansion [69]. These data support the detrimental effect of alcohol abuse on the function of dendritic cells. However, no study has characterized the effect of chronic alcohol consumption on intestinal DCs in mice and humans related to or suffering from ALD.

#### 3.3.3. Plasma Cells

The number of intestinal IgA-secreting plasma cells is reduced in experimental models of ethanol-induced liver disease in mice [70]. In humans, studies aimed at investigating IgA during ALD are scarce and contradictory. In one study, patients with decompensated cirrhosis had reduced intestinal secretion of IgA compared to compensated alcohol-associated cirrhosis [71]. By contrast, two other studies reported similar levels of IgA secretion [72] and IgA-secreting plasma cells [73]. Characterization of IgA levels with different methods as well as characterization of IgA-plasma cells at different stages of ALD and possible impact of a period of abstinence before admission may partially explain these conflicting observations. Future studies are needed to elucidate the role of IgA in the development of early and late stages of alcohol-associated liver disease. 

#### 3.3.4. T Cells

Intestinal T cells may play an important role in gut barrier dysfunction during ALD. Recent findings from our laboratory show a reduced number of T lymphocytes in the duodenum of actively drinking subjects associated with elevated microbial translocation [15]. Gene expression of Th1- and Th17-related cytokines remained unchanged in the mucosa of AUD patients compared to the controls [15]. These data rule out mucosal T cell pro-inflammatory responses as a significant driver of microbial translocation and point to a potential dysfunction in adaptive immunity. Pathophysiological mechanisms underlying the loss of intestinal T cells with identification of subsets that are specifically reduced could potentially pave the way for novel therapeutic targets directed towards improving gut barrier function. Impaired adaptive immunity linked to defective microbial immunosurveillance is a characteristic not only confined to early stages of ALD but is also common to more advanced liver disease, such as alcoholic hepatitis and cirrhosis. Reduced mucosa-associated invariant T (MAIT) cells, which play an important role against bacterial infections, have been observed in advanced ALD along with compromised antibacterial and cytotoxic responses [74]. Taken together, these data point to the contribution of different cell types to the overall dysfunctional defense mechanisms in the gut barrier during different stages of alcohol-associated liver disease (Figure 1B).

## 4. How Can Gut Barrier Dysfunction Contribute to ALD Progression?

The intestine and liver communicate via bidirectional links through the biliary tract, portal vein, and systemic circulation. The liver synthesizes and transports primary bile acids and antimicrobial molecules to the intestinal lumen through the bile duct. In the gut, the host and microbes metabolize endogenous (bile acids and amino acids) and exogenous substrates (dietary and environmental factors) into products that are then transported to the liver through the portal vein. Furthermore, the systemic circulation connects the gut-liver axis by transporting liver metabolites into the intestine [58]. Many of those products of metabolites (e.g., bile acids, fatty acids, ethanol metabolites) might act as signaling molecules at the local and distant sites where they activate various signaling processes (e.g., cell growth and differentiation, host metabolism, and immune responses) involved in regulation of the intestinal barrier [75] (Figure 2). Changes in the levels and in the structure of these signaling molecules, for example, a different proportion of secondary and primary bile acids, can negatively affect the processes (e.g., intestinal permeability, epithelial differentiation, absorption of nutrients and immunity) essential for defense mechanisms in the gut. As a result, breakdown of gut barrier function allows microbes and/or their products to get into the liver and systemic circulation, where they are recognized by different pattern recognition receptors in various cell types with subsequent activation of immune responses. According to the stage of ALD, the cell types, and signaling pathways involved, activation of a pro-inflammatory response against microbial antigens might be beneficial or detrimental for liver disease progression [76]. We here discuss some recent findings that support the link between gut barrier dysfunction, systemic inflammation, and impaired hepatic immunity in the pathogenesis of early and advanced stages of ALD.

### 4.1. Systemic Inflammatory Responses

Increased systemic translocation of bacterial products, such as lipopolysaccharides (LPS) and peptidoglycans (PGN), have been shown to be associated with elevated plasma levels of inflammatory cytokines (IL1β, IL8, IL18, TNF-α, and IL6) in AUD patients [77]. Mechanistic analyses have shown that in AUD patients, LPS and to a greater extent PGN can contribute to the activation of peripheral blood mononuclear cells (PBMCs). PBMCs are likely contributing to PGN-triggered release of IL1β, IL8, and IL18 into the blood [77]. In contrast, TNF-α and IL6 were not elevated in PBMCs, suggesting that their increased plasma levels might originate from other sources [78]. Inflammatory chemokine monocyte chemoattractant protein 1 (MCP-1) levels in the plasma are increased in actively drinking subjects compared to the controls [79]. This may promote monocyte recruitment and infiltration into the liver where they transform into inflammatory monocyte-derived macrophages. This process has been observed in mice chronically fed ethanol [80] and it is believed to play an important role in the early pathogenesis of ALD [81]. Future studies are needed to clarify the pathways linked to this process. In contrast to the early stages of alcohol-associated liver disease, severe alcohol-associated hepatitis is characterized by massive hepatic infiltration of neutrophils coming from the circulation [81]. However, a recent investigation found that alcohol-associated hepatitis patients with more neutrophils had a better outcome than those with less neutrophil infiltration [82], probably due to defective bacterial clearance and impaired liver regeneration. 

### 4.2. Impairment of Immune Responses Related to Microbes in the Liver

Increasing evidence supports the role of unbalanced immunity in the different stages of ALD. When the gut barrier is disrupted, higher amounts of microbes and/or their products translocate to the liver [55], leading to the activation of the hepatic immune system through pattern recognition receptors like Toll-like receptors (TLRs) [83]. TLRs are expressed widely in immune and non-immune cells [84] in the liver and their activation by different PAMPs coming from the intestine contributes to ALD progression [85]. 

It has been proposed that activation of the TLR4 signaling pathway by LPS plays a pivotal role in disease progression. Mice deficient in TLR4, CD14, and LPS-binding protein (LBP) show resistance to alcohol-induced liver injury [86]. Other studies in mice show the pathophysiological importance of TLR4 in resident macrophages (e.g., Kupffer cells), monocyte-derived macrophages, and hepatic stellate cells [87]. The concept of TLR4 activation with subsequent TNF-α release has conducted to translational approaches such as the anti-TNF-α therapies for patients with severe alcoholic hepatitis. The disappointing outcome of these clinical trials [88] highlight two main facts in the field of alcohol-associated liver disease: firstly, data obtained in animal models of ALD cannot be necessarily extrapolated to humans [16]; secondly, but not in order of importance, inflammatory responses can be beneficial or detrimental depending on the cell type involved and stage of liver disease. In this context, we have recently revealed that intracellular TLRs, in particular TLR7, increased at early stages of ALD in humans, whereas TLR4 did not, and was correlated with liver disease severity [79]. TLR7 upregulation occurred concomitantly with activation of interferon signaling predominantly in hepatocytes [79]. In contrast, specific TLR7 activation did not exacerbate liver disease in a chronic ethanol-feeding model [89]. 

Kupffer cells and monocyte-derived macrophages also exhibit different functions in different stages of ALD. Animal models show an important role of TLR4 activation with release of pro-inflammatory cytokines in alcohol-induced liver injury [90]. Pro-inflammatory cytokines produced by hepatic macrophages also seem to be important in the early pathogenesis of ALD [91]. However, defective macrophages in more advanced stages of liver disease may not allow optimal clearance of microbes by phagocytosis, leading to severe infections, a feature often encountered in patients with severe alcohol-associated hepatitis [92] and decompensated cirrhosis [93]. 

## 5. Conclusions and Future Directions

Gut barrier dysfunction is increasingly recognized as a main driver of alcohol-associated liver disease progression. A complex dynamic disturbance of defense mechanisms allows microbes and/or microbial antigens to translocate into the liver and systemic circulation, thus provoking immune responses contributing to liver damage. 

Several studies have revealed the dual roles of inflammatory and tolerogenic immune responses in microbial immunosurveillance in the intestine and liver according to the stage of ALD. Many pathophysiological aspects of the complex host–microbe interactions need to be dissected and future translational investigations should focus on getting a better insight into the dynamics between microbiota, intestinal, and hepatic cell types. Functional impairment of different specialized cell types and pathways contribute to the overall failure of the gut barrier and ALD progression. In the absence of effective pharmacological therapy, abstinence represents the only option for ALD treatment. We found that a short period of abstinence normalized increased intestinal permeability. However, serum levels of microbial translocation and liver cell damage markers remained significantly higher than control levels [15]. These results indicate that abstinence alone is not sufficient to improve all the modifications in the intestinal barrier linked to ALD. In this review, we have highlighted the fact that multiple cell types of the gut barrier are involved in mucosal dysfunction in a manner dependent on liver disease severity. Targeting only one aspect of alcohol-associated intestinal barrier dysfunction could not be sufficient to improve and/or reverse ALD. Therefore, multi-targeted and personalized therapy appears to be the only solution to finally treat these patients. For this purpose, standardized clinical studies are needed to stratify AUD patients according to their degree of gut barrier dysfunction, liver disease severity, and systemic inflammatory response. Identification of novel biomarkers related to these pathophysiological conditions will help clinicians to subdivide patients into subgroups linked to disease state and potentially treat them in a targeted fashion.

Innovative technological developments, such as metagenomic and metatranscriptomics, single-cell analysis, spatial transcriptomics, and mass spectrometry-based technologies, will help us to identify promising biomarkers and finally understand the complexity of the different factors and pathways involved in gut barrier dysfunction during ALD. Generation of small intestinal enteroids from primary adult stem cells and co-culture with different subsets of gut immunocytes from the same patients will help us to study the dynamics of the immune–epithelium crosstalk involved in defense mechanisms against microbes. A deep understanding of the causal links between changes in the gut microbiome and intestinal barrier dysfunction at different stages of alcohol-associated liver disease will provide us with the development of new diagnostic and prognostic tools and with the identification of new therapeutic targets. 

## Figures and Tables

**Figure 1 ijms-22-12687-f001:**
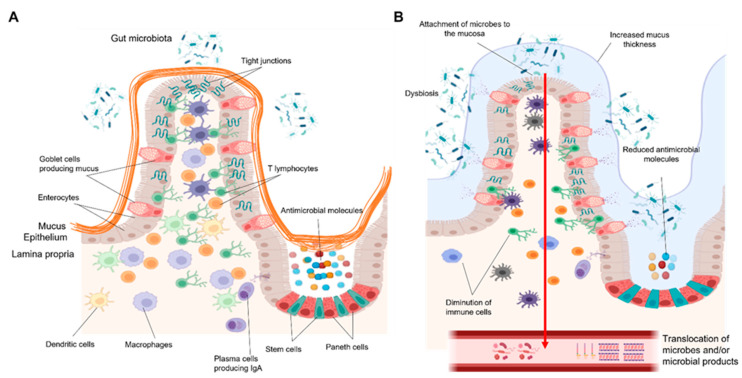
(**A**) The small intestinal barrier in healthy conditions. The intestinal barrier is composed of three main layers: the mucus, the epithelium, and the lamina propria. The epithelium, which is overlaid by a thin and discontinuous mucus layer, together with enterocytes and specialized cells constitute a first line of defense. Immune cells in the mucosa further fine-tune the defense mechanisms against pathogens. Macrophages underlying the epithelium regulate immune response by phagocytosis of microbes and production of large amounts of anti-inflammatory cytokines, thus preventing excessive immune responses. Dendritic cells capture, process, and present microbial antigens to different adaptive immune cells. T lymphocytes rapidly act against pathogens by killing infected cells, producing cytokines, and coordinating immune responses. (**B**) The small intestinal barrier in patients with an alcohol-use disorder. Chronic alcohol consumption is associated with alterations in the composition of the gut microbiota, increased attachment of microbes to the intestinal mucosa, and their translocation into the portal and systemic circulation. This process is enhanced by impairment of various epithelial and immune defense mechanisms. A loose, thickened mucus layer as well as reduced production of antimicrobial molecules and a diminution of macrophages and T lymphocytes all contribute to the failing gut barrier in AUD patients. Figures were created with Biorender.com (14 November 2021).

**Figure 2 ijms-22-12687-f002:**
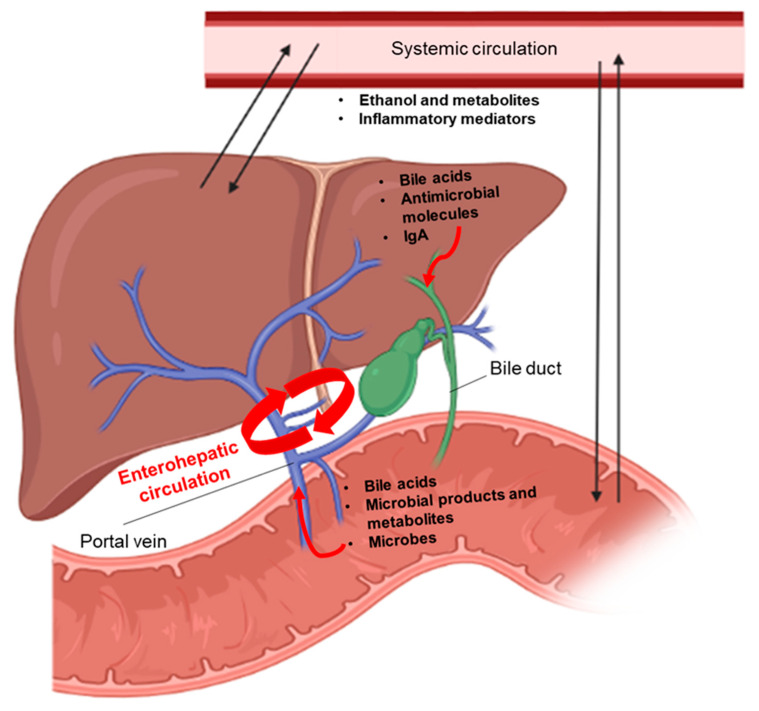
Bidirectional communication between the intestine and the liver. The liver produces primary bile acids, antimicrobial molecules, and IgA that are released in the intestine through the bile duct. In the intestine, these molecules contribute to shape the microbiota. In addition, primary bile acids are converted into secondary bile acids by the gut microbiota. Bile acids, which are reabsorbed in the terminal ileum, microbial products and metabolites, as well as viable microbes are transported to the liver through the portal vein. Once in the liver, they are implicated in triggering immune and inflammatory responses that might lead to liver disease. Moreover, the gut–liver axis is connected via the systemic circulation where ethanol, ethanol-derived metabolites, as well as other inflammatory mediators (cytokines, metabolites, etc.) can reach the two organs, thus influencing their functions. Figures were created with Biorender.com (14 November 2021).

## Data Availability

The data that support the findings of this study are available from the corresponding author upon reasonable request.

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
