# Peer review of "Host Factors in Dysregulation of the Gut Barrier Function during Alcohol-Associated Liver Disease"

_ijms, 2021, doi:10.3390/ijms222312687_

Round 1

Reviewer 1 Report

In the present manuscript Maccioni et al. describe  the Gut barrier dysfunction and its association with alcohol liver disease.

Please read my comments below:

Minor:

What is the term of subjects with an alcohol use disorder (AUD), is it a medical term?

ALD stands for Alcoholic liver disease, not alcohol-associated liver disease, in any case it would be alcohol-related liver disease (ARLD).

Major:

The introduction seems to me to be short. It does not reflect the state of the art of the problem and does not justify what is new in this review.

It is not specified whether the search has been limited to a specific date, nor what this review contributes that has not been included in others (I indicate a couple of examples below).

It should be clear whether human or animal studies or both are included. If it is the latter option, it should be clearly stated when data of one type are given and when of the other, as well as whether there are controversies in this regard.

Zhou Z, Zhong W. Targeting the gut barrier for the treatment of alcoholic liver disease. Liver Res. 2017 Dec;1(4):197-207. doi: 10.1016/j.livres.2017.12.004. PMID: 30034913; PMCID: PMC6051712.

Subramaniyan V, Chakravarthi S, Jegasothy R, Seng WY, Fuloria NK, Fuloria S, Hazarika I, Das A. Alcohol-associated liver disease: A review on its pathophysiology, diagnosis and drug therapy. Toxicol Rep. 2021 Feb 19;8:376-385. doi: 10.1016/j.toxrep.2021.02.010. PMID: 33680863; PMCID: PMC7910406.

In my opinion, the authors should re-write this work by adding novelty to what has already been contributed by other authors in order to make it attractive to the scientific community.

Author Response

We thank the reviewer for the valuables remarks and suggestions, which allowed us to further improve the review.

“In the present manuscript Maccioni et al. describe the Gut barrier dysfunction and its association with alcohol liver disease.

Major:

The introduction seems to me to be short. It does not reflect the state of the art of the problem and does not justify what is new in this review.

Response: We thank the reviewer for this important remark and we agree that the introduction could benefit from more expansion on the novelty of the findings we are reporting. We rewrote the introduction and added new data and references.

It is not specified whether the search has been limited to a specific date, nor what this review contributes that has not been included in others (I indicate a couple of examples below).

Response: We agree that it might not have been sufficiently clear what the intention of this review was. In the examples cited by the reviewer the authors reviewed how targeting specifically the gut microbiota might be a treatment option for alcoholic liver disease or described  treatment options specifically for patients with severe alcohol-associated hepatitis. Our focus does not lie on treatment but on host-related intestinal alterations in the different stages of ALD. We now emphasize the fact this review aims at describing some recent advances made in the comprehension of host factors involved in the maintenance of the intestinal barrier in healthy conditions and changes linked to alcohol abuse and ALD progression. The title has also been changed accordingly.

It should be clear whether human or animal studies or both are included. If it is the latter option, it should be clearly stated when data of one type are given and when of the other, as well as whether there are controversies in this regard.

Response: We agree that it should be specified whether data cited in the paper are derived from animal and/or human studies. We carefully checked the manuscript and indicated whenever relevant whether the data have been obtained in humans or from animal studies. We also added a paragraph in the introduction where we describe the limitations of using animal models of ALD which explains why controversies do exist and why it is difficult to extrapolate findings in animals to human pathology. We also state that we paid specific attention to include (and to identify as such) the available data described in humans in this review.

Zhou Z, Zhong W. Targeting the gut barrier for the treatment of alcoholic liver disease. Liver Res. 2017 Dec;1(4):197-207. doi: 10.1016/j.livres.2017.12.004. PMID: 30034913; PMCID: PMC6051712.

Subramaniyan V, Chakravarthi S, Jegasothy R, Seng WY, Fuloria NK, Fuloria S, Hazarika I, Das A. Alcohol-associated liver disease: A review on its pathophysiology, diagnosis and drug therapy. Toxicol Rep. 2021 Feb 19;8:376-385. doi: 10.1016/j.toxrep.2021.02.010. PMID: 33680863; PMCID: PMC7910406.

In my opinion, the authors should re-write this work by adding novelty to what has already been contributed by other authors in order to make it attractive to the scientific community.”

 “Minor:

What is the term of subjects with an alcohol use disorder (AUD), is it a medical term?

Response: Alcohol use disorder is indeed a medical term that is widely used since publication of the 5th edition of the Diagnostic and Statistical Manual of Mental Disorders (DSM). DMS defines alcohol dependence as a cluster of behavioral, cognitive and physiological phenomena that develop after repeated alcohol use and typically include a strong desire to consume alcohol, difficulties in controlling its use, persisting in its use despite harmful medical and social consequences, a higher priority given to alcohol use than other activities and obligations, increased tolerance, and sometimes a physiological withdrawal state (1). Since May 2013, the fifth edition (DSM-5) integrates the term alcohol dependence with alcohol abuse, which is a less severe psychiatric disease and which essentially insists on the consequences of drinking rather than on a specific symptomatology, into a single disorder called “alcohol use disorder”. In addition, DSM-5 added alcohol craving as a criterion for AUD diagnosis, which was not included in the prior edition (2).

  • Gilpin NW, Koob GF. Neurobiology of alcohol dependence: focus on motivational mechanisms. Alcohol Res Health. 2008;31(3):185-95. PMID: 19881886; PMCID: PMC2770186.
  • https://www.niaaa.nih.gov/publications/brochures-and-fact-sheets/alcohol-use-disorder-comparison-between-dsm

ALD stands for Alcoholic liver disease, not alcohol-associated liver disease, in any case it would be alcohol-related liver disease (ARLD).”

Response: We agree that ALD in the past had been used for Alcoholic liver disease. However, the American Association for the Study of Liver Diseases recently proposed to change this term to “Alcohol-associated liver disease” which is the reason why we used it in our review (3).

  • Crabb DW, Im GY, Szabo G, Mellinger JL, Lucey MR. Diagnosis and Treatment of Alcohol-Associated Liver Diseases: 2019 Practice Guidance From the American Association for the Study of Liver Diseases. Hepatology. 2020 Jan;71(1):306-333. doi: 10.1002/hep.30866. PMID: 31314133.

Reviewer 2 Report

Dear Authors,

After the review process, I have several comments: you should mention how the figures were realized in the figure's legend; alcohol consumption is often associated with the obesity and neurodegenerative pathologies incidence and in this case, the link between obesity, microbiota dysbiosis, and neurodegenerative pathogenesis is a new finding that should be included in the paper; the introduction is too short, the number of references is not present, only one and two short paragraphs.

Best regards.

Author Response

We thank the reviewer for the suggestions. Please find below the point-by-point answers to the different aspects.

  1. you should mention how the figures were realized in the figure's legend.

Reply: We apologize for having omitted this point from the legends. We now added the software that has been used to realize the figures to the figure legends.

  1. alcohol consumption is often associated with the obesity and neurodegenerative pathologies incidence and in this case, the link between obesity, microbiota dysbiosis, and neurodegenerative pathogenesis is a new finding that should be included in the paper

Reply: we agree with the reviewer that alcohol does not only affect the liver and that the nervous system (especially the brain and peripheral nerfs) is particularly vulnerable to alcohol-induced damage. Although it is clear that a gut-brain axis does exist, we did not include those aspects in our review. The paper has been written for a special issue dealing with Gut-liver axis and we think that including neurodegenerative findings would go too far beyond the scope of the paper.  

With regard to obesity, we agree that there might be an overlap between alcohol-induced liver damage and damage related to non-alcoholic steatohepatitis especially in obese patients with the metabolic syndrome. Although animal studies have to be taken with caution, some data indicate that there might be a synergistic action. In humans, most of the literature has investigated the relationship between both issues primarily in NASH patients consuming low or moderate amounts of alcohol with conflicting results. In general, true alcohol-dependent patients were excluded from those studies. Our intention was to principally focus on alcohol as the primary cause of liver damage and therefore we did not review potential confounding factors in detail. We now mention in the introduction that obesity might be an important factor that contributes to progression of liver disease in combination with alcohol but we do not review this aspect in more detail given the relative restricted focus of our paper. Instead, we refer the reader to two recent reviews on the subject for more reading.

  1. the introduction is too short, the number of references is not present, only one and two short paragraphs.

Reply: We re-wrote and expanded the introduction adding additional aspects as well as reasons for controversy between animal and human data. We apologize for having omitted references in the initial version. This has been corrected as appropriate.

Reviewer 3 Report

This manuscript reviews literature on the mechanisms tying intestinal epithelial barrier dysfunction and alcohol-associated liver disease. The translocation of microbes and their metabolites systemically initiates an immune response in the liver, leading to chronic and progressive tissue destruction. The authors have focused this review on host regulation of the gut barrier function, leaving discussion of microbiota-associated regulation to others. Given this choice, the manuscript likely needs to be retitled to emphasize that discussion only on the host side of the barrier is included here, given the level of interaction with the microbiome can’t really be untangled.

Major:

  1. For Figure 1, assume this should be a diagram of the small intestine given this is a discussion on portal transfer (largely) of microbes? I don’t think the authors should simply generalize by stating ‘intestine’ as there are distinct differences between small and large, and even among the portions of small itself. For example, the mucus layer is continuous and thick in the large intestine, but discontinuous and thin in the small intestine. Assuming we are detailing duodenum or small intestine here, the diagram should demonstrate thin, discontinuous mucus layer. The figure also needs to state that this is a diagram of the healthy, normal small intestine, in contrast to Figure 2.
  1. Figure 2 introduces the topic of intestinal dysbiosis, but this is not really discussed in the manuscript itself at all. Bacterial invasion/translocation is discussed, in general terms. A section on dysbiosis needs to be included if this contributes (and it likely does) to increased bacterial translocation, or influences the host intestinal epithelium in a way that leads to increased translocation, if the discussion in the Figure caption is kept as is. If the authors want to leave discussion of the microbiome to the 23rd reference, then remove the 2nd sentence from Figure 2 caption.
  1. As mentioned above, retitle manuscript to indicate that this is only a discussion of host epithelium regulation rather than also including discussion on microbiome changes or influences on the host epithelial response.
  1. Figures 1 and 2 can likely be combined into a single figure. Would suggest then adding a second figure demonstrating the path between liver and gut (portal vein, portal duct, etc.), and the products and metabolites that are transferred between the two sites.

Minor:

  1. Lines 30-31—Would suggest “immunosurveillance of microbes” rather than “microbial immunosurveillance” … the latter sounds like the microbes are the ones surveilling.
  1. Lines 85-86, what is meant by “housing the epithelium”?
  1. Line 91, define acronym GALT.
  1. Line 110, delete ‘s’ from protects.
  1. Lines 116 and 251, should be ‘synthesize’ rather than ‘synthetize’.
  1. Line 127, replace ‘to’ with ‘in’.
  1. Line 135, remove ‘g’ from ‘occluding.’
  1. Line 137, remove hyphen from ‘steato-hepatitis.’
  1. Line 145, please cite publication for human data of muc2 increase associated with alcohol…reference 27 is based only on animal models.
  1. Lines 170 and 173, remove ‘h’ from 'Cathenin'.
  1. Line 204, replace ‘exposure’ to ‘exposed.’
  1. Line 261, remove word ‘like.’

Author Response

We are grateful for the remarks and suggestions, which allowed us to clarify several aspects of the review and to further improve its content.

“This manuscript reviews literature on the mechanisms tying intestinal epithelial barrier dysfunction and alcohol-associated liver disease. The translocation of microbes and their metabolites systemically initiates an immune response in the liver, leading to chronic and progressive tissue destruction. The authors have focused this review on host regulation of the gut barrier function, leaving discussion of microbiota-associated regulation to others. Given this choice, the manuscript likely needs to be retitled to emphasize that discussion only on the host side of the barrier is included here, given the level of interaction with the microbiome can’t really be untangled.

As mentioned above, retitle manuscript to indicate that this is only a discussion of host epithelium regulation rather than also including discussion on microbiome changes or influences on the host epithelial response.”

Response: We totally agree with this comment and we have therefore retitled the manuscript as following: “Host factors in dysregulation of gut barrier function during alcohol-associated liver disease”. To emphasize the reasons behind our choice for this review, we have rewritten the introduction as following:

“Figure 2 introduces the topic of intestinal dysbiosis, but this is not really discussed in the manuscript itself at all. Bacterial invasion/translocation is discussed, in general terms. A section on dysbiosis needs to be included if this contributes (and it likely does) to increased bacterial translocation, or influences the host intestinal epithelium in a way that leads to increased translocation, if the discussion in the Figure caption is kept as is. If the authors want to leave discussion of the microbiome to the 23rd reference, then remove the 2nd sentence from Figure 2 caption.”

Response: We thank the reviewer for this important remark. We have added two sections on the gut microbiota: in the first (2.1 Gut Microbiome) we describe the role of the microbiota in the maintenance of intestinal barrier function, while in the second one (3.1 Alcohol-associated changes in the gut microbiome) we summarize the major alcohol-associated changes in the microbiota and how these alterations could influence microbial translocation or the host mucosal cells in a way that leads to microbial translocation.

.

“For Figure 1, assume this should be a diagram of the small intestine given this is a discussion on portal transfer (largely) of microbes? I don’t think the authors should simply generalize by stating ‘intestine’ as there are distinct differences between small and large, and even among the portions of small itself. For example, the mucus layer is continuous and thick in the large intestine, but discontinuous and thin in the small intestine. Assuming we are detailing duodenum or small intestine here, the diagram should demonstrate thin, discontinuous mucus layer. The figure also needs to state that this is a diagram of the healthy, normal small intestine, in contrast to Figure 2.”

Response: We thank the reviewer for this important remark. We have changed the figure accordingly showing a discontinuous and thin mucus typical of the small intestine and we have adapted the figure legend accordingly.

“Figures 1 and 2 can likely be combined into a single figure. Would suggest then adding a second figure demonstrating the path between liver and gut (portal vein, portal duct, etc.), and the products and metabolites that are transferred between the two sites.”

Response: We thank the reviewer for the suggestion that allows us to improve significantly the review. We have combined Figure 1 and Figure 2 into a single figure. We also added a new figure 2 showing the bidirectional exchange between the liver and the intestine, with the products and metabolites that are transferred between the two sites.

Figure 2. Bidirectional communication between the intestine and the liver. The liver produces primary bile acids, antimicrobial molecules and IgA that are released in the intestine through the bile duct. In the intestine, these molecules contribute to shape the microbiota. In addition, primary bile acids are converted into secondary bile acids by the gut microbiota. Bile acids which are reabsorbed in the terminal ileum, microbial products and metabolites as well as viable microbes are transported to the liver through the portal vein. Once in the liver, they are implicated in triggering immune and inflammatory responses that might lead to liver disease. Moreover, the gut-liver axis is connected via the systemic circulation where ethanol, ethanol-derived metabolites, as well as other inflammatory mediators (cytokines, metabolites etc.) can reach the two organs thus influencing their functions. Figures were created with Biorender.com.

“Minor:

  1. Lines 30-31—Would suggest “immunosurveillance of microbes” rather than “microbial immunosurveillance” … the latter sounds like the microbes are the ones surveilling.
  1. Lines 85-86, what is meant by “housing the epithelium”?
  1. Line 91, define acronym GALT.
  1. Line 110, delete ‘s’ from protects.
  1. Lines 116 and 251, should be ‘synthesize’ rather than ‘synthetize’.
  1. Line 127, replace ‘to’ with ‘in’.
  1. Line 135, remove ‘g’ from ‘occluding.’
  1. Line 137, remove hyphen from ‘steato-hepatitis.’
  1. Line 145, please cite publication for human data of muc2 increase associated with alcohol…reference 27 is based only on animal models.
  1. Lines 170 and 173, remove ‘h’ from 'Cathenin'.
  1. Line 204, replace ‘exposure’ to ‘exposed.’
  1. Line 261, remove word ‘like.’ “

Response: We corrected the manuscript accordingly for the points 1, 3, 4, 5, 6, 7, 8, 10, 11, 12.

For the point 2 we changed the text as following: “Dendritic cells (DCs) orchestrate mucosal immunity by acquiring antigens in the epithelium through goblet or microfold epithelial cells or by directly capturing and sampling luminal antigens”.

Regarding the point 9, the reference about the increased mucus thickness in mice and humans is the same:

  1. Hartmann, P. Chen, H. J. Wang, L., Wang, D. F. McCole, Brandl K. et al., “Deficiency of intestinal mucin-2 ameliorates experimental alcoholic liver disease in mice,” Hepatology, vol. 58, no. 1, pp. 108–119, 2013, doi: 10.1002/hep.26321.

Round 2

Reviewer 1 Report

The present manuscript is a considerable improvement over the previous version.

However, I still have some comments for the authors:

  • The figure captions are too long. It would be better to summarize and explain it in the text with the corresponding call to the figure.
  • Review how to indicate references when there is more than one citation for the paragraph or sentence.
  • The search range of the bibliography has not been specified and many of the references are more than 5 years old.
  • It cannot be said that this review is about recent developments.
  • Many studies included in this article are reviews. It should be changed to the original source.

Author Response

We are grateful for the remarks and suggestions.

“The figure captions are too long. It would be better to summarize and explain it in the text with the corresponding call to the figure.”

Response: We agree with this comment. We have therefore shortened the text in the figure legends removing phrases that are already explained in the main text.

“Review how to indicate references when there is more than one citation for the paragraph or sentence.”

Response: We thank the reviewer for this important remark. We have changed the format of the references.

“The search range of the bibliography has not been specified and many of the references are more than 5 years old. It cannot be said that this review is about recent developments. Many studies included in this article are reviews. It should be changed to the original source.”

Response: We thank the reviewer for this important remark. We have added additional references. With this modification in the manuscript, more than 50% of the references are now less than 5 years old. Since our primary aim was to focus on host factors, we are now stating in the introduction that “this review aims to highlight advances made in the comprehension of the host factors involved in the maintenance of the intestinal barrier in healthy conditions and changes linked to alcohol abuse and ALD progression with specific attention paid to data described in humans” omitting the word “recent” as suggested. In addition, we have added  original articles to the cited reviews (citation 37, 40, 41).

Reviewer 2 Report

Dear Authors,

You do not respond to the main associated pathology - obesity. You should read again my first set of comments and make the necessary modifications in the paper. In the introduction, you should add new references (part of them are too old) to sustain the dysbiosis and that explains the connection with obesity. Best regards.

Author Response

You do not respond to the main associated pathology - obesity. You should read again my first set of comments and make the necessary modifications in the paper. In the introduction, you should add new references (part of them are too old) to sustain the dysbiosis and that explains the connection with obesity.”

Response: We thank the reviewer for this remark. We acknowledged the relation between obesity and alcohol consumption. However, looking into the studies in more detail, most of them are dealing with alcohol consumption in pre-existing obesity or are based on animal NAFLD models where alcohol has been added on. We also agree that dysbiosis has been shown in obese subjects and that alcohol might aggravate the changes in the microbiota in those patients. We are now discussing these aspects in more detail in the introduction and we have also added recent references that potentially sustain a synergism between obesity and alcohol consumption.

“The interaction between obesity and alcohol, in particular for mild or moderate consumption, is controversial in humans[4]. The incidence of obesity and the associated metabolic syndrome is increasing worldwide. A recent cohort-study showed that patients with excessive alcohol consumption have increased prevalence of obesity and metabolic syndrome[5] which might increase the risk for liver disease progression. It has been reported that moderate alcohol consumption increases the risk of development of hepatic steatosis in association with obesity. By contrast, moderate alcohol consumption reduces mortality in normal body mass index (BMI) and overweight individuals. This beneficial effect was, however, lost in obese subjects [6]. In rodents, the synergism between alcohol and obesity might be linked to alterations of immune responses, metabolism and gut microbiota composition[3]. Recent animal data where alcohol binges have been added on top of a non-alcoholic steatohepatitis model support a synergistic effect for liver damage (for detailed review see ref.[7]).”

[3]         F. M. F. Åberg and Martti Färkkilä , “Drinking and Obesity : Alcoholic Liver Disease / Nonalcoholic Fatty Liver Disease Interactions,” Semin. Liver Dis., pp. 154–162, 2020.

[4]         G. Traversy and J. P. Chaput, “Alcohol Consumption and Obesity: An Update,” Curr. Obes. Rep., vol. 4, no. 1, pp. 122–130, 2015.

[5]    A. Singh, H. Amin, R. Garg, M. Gupta, R. Lopez, N. Alkhouri N et al., Increased Prevalence of Obesity and Metabolic Syndrome in Patients with Alcoholic Fatty Liver Disease. Dig Dis Sci. 2020 Nov;65(11):3341-3349.

[6]    T. B. Peeraphatdit, J. C. Ahn, D. H. Choi, A. M. Allen, D.A. Simonetto, P. S. Kamath P et al., A Cohort Study Examining the Interaction of Alcohol Consumption and Obesity in Hepatic Steatosis and Mortality. Mayo Clin Proc. 2020 Dec;95(12):2612-2620.

[7]         S. Hwang, T. Ren, and B. Gao, “Obesity and binge alcohol intake are deadly combination to induce steatohepatitis: A model of high-fat diet and binge ethanol intake,” Clin. Mol. Hepatol., vol. 26, no. 4, pp. 586–594, 2020.

Reviewer 3 Report

Much improved, thank you!

Author Response

Thank for your suggestions and support.

Round 3

Reviewer 1 Report

Next time, please underline all modifications that you describe in the text of letter as a response to the reviewer also in the text of the manuscriptplace where they have  bein done.